# Chimeric Peptide Engineered Nanomedicine for Synergistic Suppression of Tumor Growth and Therapy-Induced Hyperlipidemia by mTOR and PCSK9 Inhibition

**DOI:** 10.3390/pharmaceutics15102377

**Published:** 2023-09-23

**Authors:** Hua Cai, Rongrong Zheng, Ningxia Wu, Jiaman Hu, Ruixin Wang, Jianing Chi, Wei Zhang, Linping Zhao, Hong Cheng, Ali Chen, Shiying Li, Lin Xu

**Affiliations:** 1Department of Geriatric Cardiology, General Hospital of Southern Theater Command, People’s Liberation Army, Guangzhou 510010, China; 20211111490@stu.gzucm.edu.cn (H.C.); wunx@stu.gzucm.edu.cn (N.W.); hujiaman327@gmail.com (J.H.); wangruixin556@126.com (R.W.); c15829881681@163.com (J.C.); 2Graduate School, Guangzhou University of Chinese Medicine, Guangzhou 510006, China; 3Guangdong Provincial Key Laboratory of Molecular Target & Clinical Pharmacology, The NMPA and State Key Laboratory of Respiratory Disease, School of Pharmaceutical Sciences and the Fifth Affiliated Hospital, Guangzhou Medical University, Guangzhou 511436, China; 2023390075@gzhmu.edu.cn (R.Z.); zlp2022@gzhmu.edu.cn (L.Z.); 4School of Chemistry and Chemical Engineering, Guangdong Pharmaceutical University, Guangzhou 510006, China; zhangweiv229@163.com (W.Z.); chenali2004@163.com (A.C.); 5Biomaterials Research Center, School of Biomedical Engineering, Southern Medical University, Guangzhou 510515, China; chengh@smu.edu.cn

**Keywords:** anti-tumor, hyperlipidemia, tumor-targeted peptide, nanomedicine, Rapamycin, proprotein convertase subtilisin/kexin type 9 (PCSK9)

## Abstract

Chemotherapy-induced side effects restrain anti-tumor efficiency, with hyperlipidemia being the most common accompanying disease to cause treatment failure. In this work, a chimeric peptide-engineered nanomedicine (designated as PRS) was fabricated for the synergistic suppression of tumor growth and therapy-induced hyperlipidemia. Within this nanomedicine, the tumor matrix-targeting peptide palmitic-K(palmitic)CREKA can self-assemble into a nano-micelle to encapsulate Rapamycin (mTOR inhibitor) and SBC-115076 (PCSK9 inhibitor). This PRS nanomedicine exhibits a uniform nano-distribution with good stability which enhances intracellular drug delivery and tumor-targeting delivery. Also, PRS was found to synergistically inhibit tumor cell proliferation by interrupting the mTOR pathway and reducing Rapamycin-induced hyperlipidemia by increasing the production of LDLR. In vitro and in vivo results demonstrate the superiority of PRS for systematic suppression of tumor growth and the reduction of hyperlipidemia without initiating any other toxic side effects. This work proposes a sophisticated strategy to inhibit tumor growth and also provides new insights for cooperative management of chemotherapy-induced side effects.

## 1. Introduction

Tumors and cardiovascular diseases have the highest morbidity and mortality rates in the world. Both of them have some common risk factors such as hyperlipidemia and diabetes, resulting in a large number of patients with malignancy accompanied by cardiovascular disease [1,2]. Of note, chemotherapeutic agents of tumors can induce cardiotoxicity with the prolongation of treatment time and patient survival, including myocardial insufficiency, heart failure, coronary heart disease, arrhythmias, hypertension, and hyperlipidemia [3]. Chemotherapy-induced cardiotoxicity can not only restrain tumor treatment efficiency, but it can also contribute as an accompanying risk factor for treatment failure. Chemotherapy-induced cardiotoxicity has become the second leading cause of death in patients with tumors other than recurrent metastases [4,5,6]. Even so, chemotherapy-induced cardiovascular disease has not received enough attention, and there is no effective strategy for the systematical management of tumors combined with cardiovascular disease.

As we know, the AKT/mTOR pathway is activated to promote cell growth, proliferation, movement, survival, apoptosis, autophagy, and angiogenesis [7]. Rapamycin is widely regarded as an effective inhibitor of AKT/mTOR, which has been approved for the treatment of neuroendocrine tumors, breast cancer, kidney cancer, prostate intraepithelial neoplasia, et al. [8,9,10,11,12]. However, some recent findings suggest that hyperlipidemia happens after treatment with Rapamycin (7–73%) [13,14,15,16,17,18]. In particular, triglyceride (TG) levels increased significantly after treatment with mTOR inhibitors [19,20]. TG was found to deposit in the arteries, increasing blood viscosity and promoting arterial sclerosis [21,22]. Even worse, hyperlipidemia could also contribute to the development of malignant tumors [23]. Therefore, there is an urgent need to develop a combination strategy to overcome mTOR inhibitors-induced hyperlipidemia during tumor treatment.

Randomized drug distribution within normal organs contributes largely to chemotherapy-induced side effects. Typically, a nanomedicine could improve the physicochemical properties and pharmacokinetics of drugs to enhance their delivery to tumors and reduce undesired side effects in normal tissues. Even so, most of the nanomedicine would be captured and excreted by the reticuloendothelial system, thereby inducing drug accumulation in the liver. Serum cholesterol plays an especially dominant role in hyperlipidemia, which is carried by low-density lipoprotein (LDL) and metabolized in the liver [24]. It should be noted that the proprotein convertase subtilisin/kexin type 9 (PCSK9) is a key regulator of LDL metabolism, which can remove the LDL by the upgradation of its receptor (LDLR) to regulate cholesterol levels [25]. Extensive evidence has confirmed that PCSK9 inhibitors could effectively reduce low-density lipoprotein cholesterol (LDL-C) in patients with a higher risk of arteriosclerotic cardiovascular disease (ASCVD) to achieve therapeutic benefits [26,27,28]. Consequently, co-delivery of PCSK9 and mTOR inhibitors may provide a promising strategy for avoiding tumor chemotherapy-induced hyperlipidemia, but such a combination is rarely reported.

In this work, a chimeric peptide engineered nanomedicine (designated as PRS) was fabricated for the synergistic suppression of tumor growth and therapy-induced hyperlipidemia. PRS was composed of a tumor matrix targeting peptide of palmitic-K(palmitic)CREKA, Rapamycin (mTOR inhibitor), and SBC-115076 (PCSK9 inhibitor), which could self-assemble into a uniform nanomedicine with good stability (Figure 1A). PRS could enhance the intracellular drug delivery and tumor-targeting delivery, synergistically inhibiting tumor cell proliferation by interrupting the mTOR pathway (Figure 1B). Meanwhile, the tumor-targeting delivery of PRS could not only reduce the systemic drug exposure-induced side effects, but also inhibit Rapamycin-induced hyperlipidemia by increasing the production of LDLR. In vitro and in vivo results demonstrated the superiority of PRS for the systematic suppression of tumor growth and reduction of hyperlipidemia without initiating any other toxic side effects. This work provided a sophisticated strategy for inhibiting tumor growth and cooperatively managing chemotherapy-induced side effects.

## 2. Materials and Methods

### 2.1. Materials and Instrumentation

Fmoc-Ala-OH, Fmoc-Lys(Boc)-OH, Fmoc-Glu(OtBu)-OH, Fmoc-Arg(Pbf)-OH, Fmoc-Cys(Trt)-OH, Fmoc-Lys(Fmoc)-OH, diisopropylethylamine (DIEA), O-benzotriazol-N, N, N′, N′-tetramethyluronium hexafluorophosphate (HBTU), and 2-chlorotrityl-chloride resin were of analytical grade and purchased from GL Biochem Ltd. (Shanghai, China). Trifluoroacetic acid (TFA), palmitic acid, ninhydrin, and triisopropylsilane (TIS) were provided by Aladdin Reagent Co., Ltd. (Shanghai, China). Dimethyl sulfoxide (DMSO) and N, N′-dimethylformamide (DMF) were obtained from Shanghai Reagent Chemical Co., Ltd (Shanghai, China).

Dulbecco’s Modified Eagle Medium (DMEM), penicillin-streptomycin, Dulbecco’s phosphate-buffered saline (PBS), and Hoechst 33342 were supplied by Invitrogen Crop. (California, USA).Fetal bovine serum (FBS) was purchased from Biological Industries Ltd. (Kibbutz Beit Haemek, Israel). The mTOR inhibitor of Rapamycin (Cat. No. HY-10219) and the PCSK9 inhibitor of SBC-115076 (Cat. No. HY-12402) were purchased from MedChem Express (Junction, NJ, USA). Sulfo-Cy5.5 (Cat. No. #GY1056) was purchased from Goyoo Biotech Co. Ltd. (Nanjing, China), Methylthiazolydiphenyl-tetrazolium bromide (MTT) was obtained from Beyotime Biotechnology Co., Ltd. (Shanghai, China). Calcein-AM/PI double stain kit was provided by Yeasen Biotech. Co., Ltd. (Shanghai, China).

The molecular weight was determined by electrospray ionization mass spectrometry (ESI-MS, ThermoFisher Scientific, Waltham, MA, USA). The particle size and polydispersity index (PDI) were measured by Zetasizer Nano ZS (Malvern, UK). The morphology of nanomedicine was observed by transmission electron microscopy (TEM, JEM-1400 PLUS). Fluorescence microscopy images in research were detected and analyzed by confocal laser scanning microscope (CLSM, ELYPA P.1, Carl Zeiss, Oberkochen, Germany). Flow cytometry analysis was conducted by amnis (ImageStreamX Mark, Merck millipore, Burlington, MA, USA). The cell viability was assessed by MTT assay with microplate reader (Mithras2 LB 943, Berthold). Inverted microscope was used to observe hematoxylin and eosin (H&E) staining. The content of Rapamycin and SBC-115076 were calculated by high-performance liquid chromatography (HPLC, LC-20AT). Blood biochemical tests were conducted by automated chemistry analyzer (Chemray-240).

### 2.2. Synthesis of Tumor-Targeted Peptide

The tumor-targeted peptide of palmitic-K(palmitic)CREKA was synthesized using the standard solid phase peptide synthesis method on 2-chlorotriacyl chloride resin. Fmoc-protected amino acid monomers were adsorbed into the resin with HBTU/DIEA as coupling agents. Piperidine/DMF (20%, *V*/*V*) was used for removal of the Fmoc protective group. A mixture of 95% trifluoroacetic acid, 2.5% triisopropylsilane, and 2.5% water was used as a cutting agent to cleave the tumor-targeted peptide from the resin. The obtained solid powder was washed by ether and then dried overnight in a vacuum. Palmitic-K(palmitic)CREKA (25 mg/mL) was dissolved in ultra-pure water with ultrasonication to form self-assembled P nanomedicine. The molecular weight was tested by ESI-MS. The particle size and zeta potential were detected by Zetasizer Nano ZS and monitored for 5 days to assess the stability.

### 2.3. Preparation and Characterization of PRS

PRS was formed by simultaneously encapsulating Rapamycin and SBC-115076 in P nanomedicine. Firstly, 10 mg/mL of Rapamycin and SBC-115076 in DMSO were dispersed into ultra-pure water and disposed of with ultrasonication for 50 s. Afterward, 25 mg/mL of palmitic-K(palmitic)CREKA was added to the above solution and the mixture was stirred with ultrasonication for 5 min. Then, the resultant mixtures were put on dialysis (MWCO 1000 Da) to eliminate the rest of the free drugs. The preparations of different molar ratios of Rapamycin and SBC-115076 were dealt with in the same way. Eventually, morphologies of PRS nanomedicines were captured by TEM, and the size was recorded by Zetasizer Nano ZS for 5 days. Rapamycin and SBC-115076 loading amounts were respectively determined by HPLC to make a standard calibration curve. The mobile phase consisted of water with 0.2% trifluoroacetic acid as solvent A and acetonitrile as solvent B. Cy5.5-PRS was prepared as follows: Rapamycin (10 mg/mL), SBC-115076 (10 mg/mL), and Cy5.5 (5 mg/mL) in DMSO were dispersed into ultra-pure water and disposed of with ultrasonication for 50 s. Next, 25 mg/mL of palmitic-K(palmitic)CREKA was added to the above solution and the mixture was stirred with ultrasonication for 5 min. Lastly, the resultant mixtures were put on dialysis (MWCO 1000 Da) to eliminate the rest of the free drugs.

### 2.4. Cell Culture and Cellular Internalization

The cell lines used in this study were derived from the Chinese Academy of Sciences Cell Bank of Type Culture Collection (Shanghai, China).. Murine breast cancer (4T1) cells were cultivated in the DMEM medium with 10% FBS and 1% antibiotics at 37 °C in a humidified incubator containing 5% CO_2_. First, Cy5.5-PRS was PRS nanomedicine labeled with the near-infrared fluorescent dye, Sulfo-Cy5.5. In addition, the cells were treated with Rapamycin (10 mg/L) in the equivalent amount of Cy5.5-PRS for 6, 12, and 24 h separately. Furthermore, 4T1 cells were respectively incubated with Rapamycin (5 mg/L, 10 mg/L, 15 mg/L) in the equivalent amount of Cy5.5-PRS for 24 h. Finally, after being stained with Hoechst 33342, the cells were observed with CLSM.

### 2.5. MTT Assay

The 4T1 cells were seeded in a 96-well plate, grown for 24 h, and then incubated with gradient concentrations of PRS for 24 h. Next, the original 200 μL DMEM per well was retained. An amount of 20 μL of 0.5 mg/mL MTT solution was directly added to each well for 4 h. Following removal of the medium and MTT solution, 150 μL of DMSO was added to each well to dissolve the purple crystal. In the end, the absorbance was measured using a microplate reader at 570 nm.

### 2.6. Live/Dead Cell Staining and Cell Apoptosis Analysis

The 4T1 cells were seeded in confocal dishes for 24 h and then incubated with Rapamycin (15 mg/L) in the equivalent amount of PRS. After 24 h, the cells were disposed of with a Calcein-AM/PI double stain kit using CLSM to observe. When testing for apoptosis, the cells were treated in a similar way as live/dead cell staining. The cells were washed with PBS three times and then stained with Annexin V-FITC/PI. Lastly, the cells in each group were collected respectively for analysis of cell apoptosis by flow cytometry.

### 2.7. Western Blot

The 4T1 cells were seeded in 6-well plates for 24 h and then incubated with Rapamycin (15 mg/L) in the equivalent amount of PRS. After 24 h, the cells were washed three times with PBS. An icy RIPA solution containing protease inhibitors and phosphatase inhibitors was added to the cells for cell cleavage (20 min). The cell cleavage fluid was then collected and broken by sonic oscillator 3 times to interrupt the deoxyribonucleic acid (DNA) and then centrifuged (12,000 rpm, 30 min, 4 °C) to remove fragmented cells, nuclei, and fragments of cells. The isolated protein was quantified with a BCA protein assay kit and normalized to the same concentration. The sample was electrophoresis separated with sodium dodecyl sulfate-polyacrylamide gel electrophoresis (SDS-PAGE) gel and transferred to the PVDF membrane, which was then closed for 1 h in a TBST containing 5% skimmed milk. The PVDF film was incubated overnight at 4 °C with the corresponding antibody. Next, the PVDF film was incubated by goat anti-rabbit or anti-mouse two resistance at room temperature for 2 h and then washed by tris-buffered saline with tween 20 (TBST) three times. Finally, the luminescent liquid was added to the PVDF film and placed into the exposure machine to detect the strip.

### 2.8. Fluorescence Imaging In Vivo

The 4T1 tumor cells (2 × 10^5^) were injected subcutaneously into BALB/c mice to construct a tumor-bearing mouse model. When the average tumor volume reached about 100 mm^3^, the mice were intravenously injected with Cy5.5-PRS. The whole-body fluorescence images were collected by real-time imaging with the in vivo fluorescence imaging system at predetermined time intervals. Additionally, the tumors and organs were harvested for fluorescence analysis. All animal experiments were conducted in accordance with the guidelines of the Institutional Animal Care and Use Committee (IACUC) of the Guangzhou Medical University (China) Animal Experimentation Center and operated in compliance with the Regulations for the Administration of Affairs Concerning Experimental Animals.

### 2.9. Tumor Growth Inhibition

The tumor-bearing mice were randomly divided into six groups (Blank, P, PS, PR, R + S, and PRS) and disposed of with each sample at a predetermined time point. In detail, on the first, third, fifth, seventh, and ninth days, the mice were intravenously injected with PRS (5.15 mg/kg), PS (1.57 mg/kg), PR (4.59 mg/kg), P (1 mg/kg), and R + S (4.15 mg/kg). The body weight and tumor volume of each mouse was recorded every 2 days and changes in the longest and shortest diameters of tumors in the mice were measured to monitor changes in tumor volume. Tumor volume was calculated as (tumor width)^2^ × (tumor length)/2.

On the twelfth day of drug injection, blood was taken from each mouse for blood biochemical tests under anesthesia. The venous blood sample was collected from the external jugular vein. Biochemical analysis included aspartate aminotransferase (AST), alanine transaminase (ALT), urea, and uric acid (UA), which were determined using an automated chemistry analyzer. Regular quality control procedures were followed and the analyzers were managed in accordance with the manufacturers’ instructions. Then, the mice were sacrificed to strip the tumor as well as record the weight of the tumor and photograph the size of tumor. The tumor and major organs were also sampled for H&E staining and imaged for fluorescence analysis.

## 3. Results

### 3.1. Synthesis and Characterization of PRS Nanomedicine

The synthesis of palmitic-K(palmitic)CREKA was prepared via the standard solid phase peptide synthesis method [29,30]. The tumor-targeted peptide of palmitic-K(palmitic)CREKA had a molecular weight of about 1210.81 Da, which was confirmed by ESI-MS (Appendix A). The P nanomedicine was prepared with 25 mg/mL of palmitic-K(palmitic)CREKA with a stable dispersion and size distribution (Figure 1A–C), which was selected for the subsequent study. The P nanomedicine, Rapamycin, and SBC-115076 were used to form the tumor-targeted PRS nanomedicine. To obtain the best nanomedicine, the assembly behavior was investigated by adjusting the feed ratios of Rapamycin and SBC-115076. Surprisingly, only at the molar ratio of 3:5 did PRS exhibit a remarkable particle size distribution and outstanding dispersibility. The morphology and size of PRS were revealed by TEM (Figure 1D) and dynamic light scattering (DLS) (Figure 1E). The changes of hydrodynamic size and PDI (Figure 1F) were monitored for 5 days, implying the considerable stability of PRS. P nanomedicine and Rapamycin, P nanomedicine and SBC-115076, respectively, were chosen to prepare PR and PS nanomedicines in the same way. A free Rapamycin and SBC-115076 mixture called R + S was also used as a control group. Interestingly, the morphology of PS was wormlike (Figure 1G), which resembled the results of the previous study [31]. Additionally, PS also had a proper size distribution (Figure 1H) and remained stable for 5 days (Figure 1I).

To measure the constitution of the nanomedicine, corresponding standard curves of Rapamycin and SBC-115076 were obtained by HPLC (Appendix A). The drug contents of Rapamycin and SBC-115076 in the nanomedicine were determined to be 7.93% and 1.25%. On the basis of the above, it could be concluded that Rapamycin and SBC-115076 could be successfully encapsulated in P nanomedicine to form PRS.

An optimum tumor-targeted nanomedicine should have cell uptake ability in order to take effect. To estimate the cellular uptake capability, CLSM was used for capturing 4T1 cells after incubation with Cy5.5-PRS. As shown in Figure 1J, red fluorescence in cells increased over time, suggesting that the uptake of Cy5.5-PRS was associated with the cultivation time in 4T1 cells. Next, after the 4T1 cells were incubated with various concentrations of Cy5.5-PRS for 24 h, the intensity of the intracellular fluorescence increased significantly as the concentration increased (Figure 1K). Arithmetic mean intensity analysis demonstrated similar results (Appendix A), indicating good self-delivery efficiency at the cellular level.

### 3.2. Cytotoxicity of PRS In Vitro

To estimate the overall effect of PRS on tumor inhibition in vitro, the 4T1 cells’ viability first was examined after treatment with gradient concentrations of P, PR, PS, R + S, and PRS. When compared with the PRS group, the MTT assay results showed that other therapeutic drugs in control groups were low in toxicity and cell survival was always higher than 90%. However, the cell proliferation capacity of the PRS group was evidently reduced, showing concentration-dependent cytotoxicity (Figure 2A). Secondly, to visualize the anti-tumor effect of nanomedicines, the live/dead cell co-staining assay was conducted via CLSM. Living cells were green fluorescent labeled Calcein-AM, while dead cells were red fluorescent labeled PI (Figure 2B). Compared with other groups, the PRS group had the strongest red fluorescence in 4T1 cells, suggesting the best anti-tumor effect (Appendix A). It was shown that the PRS group had significantly high cytotoxicity, but only R and only S-treated groups (like PR, PS, and R + S groups) were shown to have low cytotoxicity. This could be linked to Rapamycin and SBC-115076, two drugs used for synergy. Considering that Rapamycin could play an anti-tumor role through apoptosis, the apoptosis status of tumor cells after treatment was tested by flow cytometry (Figure 2C). It was difficult to induce apoptosis with the PR or PS groups alone. However, compared with the PR and PS groups, the R + S and PRS groups had improved the therapeutic effect, which deeply proved that the combination of Rapamycin and SBC-115076 might achieve a synergistic anti-tumor effect. Moreover, the anti-tumor effect of the PRS group was especially obvious, greatly implying the advantages of this self-assembly strategy in drug delivery and synergistic therapy. To explore whether apoptosis caused by PRS was associated with the AKT/mTOR pathway, a western blot assay was conducted (Figure 2D). As shown in Figure 2E, each group of Rapamycin was normalized. The results showed that the groups with Rapamycin (like PR, R + S, and PRS groups) could inhibit the AKT/mTOR pathway, speculating that nanomaterials might induce apoptosis via the AKT/mTOR pathway.

### 3.3. Anti-Tumor Study In Vivo

The anti-tumor effect of PRS was further confirmed by establishing a 4T1 tumor-bearing mice model. On the first, third, fifth, seventh and ninth days, the mice were administered nanotherapeutic drugs via intravenous injection (Figure 3A). Cy5.5-PRS was injected via tail vein into the primary tumor model on the BALB/c mice before using real-time fluorescence imaging prior to this, in order to investigate the biodistribution of PRS. As illustrated in Figure 3B, Cy5.5-PRS still maintained higher fluorescence intensity in the tumor site, indicating that it had a long retention capacity in tumors. After 24 h, the tumors and major organs of the mice were separated for fluorescence imaging. Cy5.5-PRS had an especially strong fluorescence in tumor tissue compared to other organs, manifesting its excellent tumor preferential aggregation ability, which might be related to active targeting of the tumor-homing peptide CREKA and the enhanced permeability and retention (EPR) effect.

After treatment for 12 days, the tumors injected with P, PR, PS, and R + S grew rapidly, revealing that the anti-tumor treatment was ineffective. In contrast, the PRS group significantly showed inhibition of tumor growth (Figure 3C–E). Moreover, the tumor tissues were sliced and used for terminal deoxynucleotidyl transferase-mediated dUTP-biotin nick end labeling (Tunel), nucleus-related antigen (Ki67), and H&E staining analyses (Figure 3F). The Tunel and Ki67 immunofluorescence was quantitatively analyzed (Appendix A). As expected, the PRS group had the strongest green fluorescence in Tunel signal and the weakest green fluorescence in Ki67 signal, meaning that the PRS could maximize the apoptosis and inhibit the proliferation of tumor cells. What’s more, H&E staining analyses also confirmed that tumor cells exhibited extensive and marked necrosis after being treated with PRS. These results showed that when used alone, mTOR inhibitors or PCSK9 inhibitors had limited effects on tumor suppression, but when combined, the tumor growth was obviously inhibited, indicating that the combination of the two drugs had a synergistic anti-tumor effect.

Because intravenous drugs inevitably gathered nonspecifically in normal tissues, it was necessary to assess the biosafety of anti-tumor drugs in clinical applications. During treatment, the mice of each group did not decrease significantly in weight, indicating that the drug’s systemic toxicity was low (Figure 4A). To assess the biosafety of PRS, blood samples from different treatment groups were extracted for biochemical analysis. Biochemical indexes of aspartate aminotransferase (AST), alanine transaminase (ALT), urea, and uric acid (UA) remained at normal levels without significant statistical differences, suggesting that liver and kidney function had not been impaired (Figure 4B–E). Meanwhile, as shown in Figure 4F, H&E staining of major organ tissues such as the heart, liver, spleen, and kidneys did not show significant histological abnormalities. However, compared with the R + S and PRS groups, the Blank, P, PR, and PS groups showed infiltration of the inflammatory cells, resulting in structural damage to the lungs. The reason might be that the combination of Rapamycin and the PCSK9 inhibitor could protect against the effects of pulmonary injury. Ruiliang Chu and his colleagues found that infiltration of inflammatory cells and hemorrhage in the lung tissue was alleviated after treatment with Rapamycin [32]. It was observed that the PCSK9 monoclonal antibody could reduce hypoxia-induced pulmonary vascular remodeling, which was associated with the inhibition of inflammation [33]. In general, these results suggested that the PRS had a good biosafety in vivo, providing the prospect of practical application in tumor therapy.

### 3.4. Effect of PRS on LDLR Expression In Vivo

Since the liver is the main site of lipid metabolism, abnormal lipid metabolism is closely related to LDLR expression in the liver. In order to investigate the effect of PCSK9 inhibitors on mTOR inhibitors-induced hyperlipidemia, the liver of the mice was collected for analysis on the twelfth day after the first drug treatment. LDLR protein expression levels were detected in histological sections from 4T1 tumor-bearing mice by immunohistochemistry (IHC). Compared with other groups, the LDLR protein levels were increased significantly in the PRS group and had significant statistical differences (Figure 5A,B). The results showed that LDLR protein levels decreased in the Blank, P, PR, PS, and R + S groups, which indicated abnormal lipid metabolism possibly caused by tumors. More importantly, compared with the Blank group, LDLR levels were reduced more significantly in the PR and R + S groups, indicating that Rapamycin could induce a decrease in LDL.

## 4. Discussion

mTOR is the mammalian target of Rapamycin and also an important serine-threonine protein kinase downstream in the PI3K/Akt pathway [34]. The previous study reported that there was an abnormal PI3K/Akt/mTOR signaling pathway in a variety of malignant tumors [35]. Sun L et al. found that mTOR inhibitors could not only inhibit the known target mTOR, but also affect the STAT3 and c-myc targets, which could inhibit three cancer genes simultaneously [36]. Taken together, the small molecule inhibitor PI3K/Akt/mTOR signaling pathway has evolved into an emerging anti-tumor drug. Rapamycin was widely considered as an effective mTOR inhibitor. Besides, PCSK9 was the ninth member of the family of proprotein convertases. In recent years, PCSK9 has aroused extensive attention in tumor progression. It could not only promote tumor development by regulating apoptosis and the invasion of tumor cells [37,38,39], but could also be involved in the regulation of tumor microenvironments by impacting the infiltration and function of immune cells and macrophages [40,41,42,43,44]. Gradually, PCSK9 has become the target of combined chemotherapy or immunotherapy and other anti-tumor therapies. To the best of our knowledge, this was the first study to explore the combination of the mTOR inhibitor and PCSK9 inhibitor in tumor therapy. We found that PRS could promote apoptosis of 4T1 tumor cells via the AKT/mTOR pathway, inhibiting breast cancer growth.

However, it has previously been reported that there is a close link between mTOR, MAPK, and PCSK9. Therefore, we also suspected that there were three other possible reasons for suppressing tumor growth in the combination of mTOR inhibitors and PCSK9 inhibitors which would be explored in greater detail in our future studies. First, PCSK9 inhibitors might remedy the disadvantage of mTOR inhibitor-induced activation of the MAPK pathway. Abnormal or excessive activation of the MAPK signaling pathway played an important role in the transformation and evolution of malignant cells [45]. Intriguingly, there was considerable crosstalk between the AKT/mTOR and MAPK signaling pathways so that inhibition of one pathway could still lead to the maintenance of signaling through the other pathway [46,47,48]. Osamu Nakamura et al. demonstrated that mTOR inhibitors induced autophagy in malignant fibrous histiocytoma cells by activating the MEK/ERK signaling pathway, and the mTOR inhibitor-induced apoptosis could be enhanced by MAPK inhibitors [49]. Similarly, Lasithiotakis KG et al. reported that the combination of Sorafenib with Rapamycin induced melanoma cell death by simultaneously and efficiently inhibiting the MAPK and mTOR signaling pathways [50]. This indicated that dual inhibition of the mTOR and MAPK parallel signaling pathways prevented compensatory activation of the redundant pro-survival pathway [51,52]. Interestingly, it was reported that PCSK9 inhibitors could decrease tumor metastasis in tumors by inhibiting the MAPK pathway through HSP70 up-regulation [43].

Second, down-regulation of PCSK9 could directly decrease levels of p-PI3K and p-AKT proteins to improve the inhibition of mTOR. Wang L et al. first revealed that the knockdown of PCSK9 expression could reduce the process of tumor cell epithelial-mesenchymal transition and depress the activation of PI3K/AKT signaling, as well as induce M1 macrophage polarization, thus suppressing tumor cell proliferation, migration, invasion, and epithelial-mesenchymal transition in vitro and tumor cell metastasis in vivo [53].

Third, the combination of PCSK9 inhibitors and Rapamycin could enhance the inhibition of the STAT3 signaling pathway, promote apoptosis, and reduce the potential proliferation and metastatic ability of malignant tumors. STAT3, one of the most activated transcription factors in cancer and gene therapy, has become an important target in malignancy treatment [36]. The transcriptional activity in STAT3 was also inhibited by Rapamycin [54]. Moreover, previous studies revealed that the inhibition of PCSK9 might have an oncogenic role in the development and progression of colorectal cancer by inhibiting JAK2/STAT3 signaling [55]. Consequently, studies of the underlying mechanisms of the combination of mTOR and PCSK9 inhibitors in tumor treatment remain to be conducted in the future.

Lipids are also one of the three major nutrients and key active molecules in cell life activities. Abnormal lipid metabolism in tumor cells is characterized by the uncontrolled synthesis of fatty acids from scratch and enhanced lipid synthesis, providing a corresponding microenvironment for tumor cell proliferation [47]. In addition to tumor disease itself, anti-tumor therapy could also result in abnormal blood lipids. mTOR plays a key role in fat metabolism and participates in regulating lipid storage in adipocytes [56]. The use of mTOR inhibitors in humans resulted in hyperlipidemia [16]. A total of 15% of patients developed hyperlipidemia in a phase I/II trial of Temsirolimus combined with Interferon Alfa for advanced renal cell carcinoma [15]. Ezra E W Cohen and his colleague recruited 138 cancer patients and found that 43% of those treated with mTOR inhibitors developed hyperlipidemia [14]. On the other hand, hyperlipidemia was one of the independent risk factors for cardiovascular disease and many researchers reported that prevention and treatment of hyperlipidemia could significantly reduce cardiovascular morbidity and mortality [57,58]. It was reported that inhibition of PCSK9 could be an effective method for reducing the unfavorable side effects of mTOR inhibitors-induced hyperlipidemia. There was evidence that the inhibition of mTOR inhibitors led to an increase of PCSK9 expression by regulating PKCδ, HNF4α, and HNF1α, a decrease of hepatic LDLR protein levels, and an increase of VLDL/LDL cholesterol levels in mice, ultimately leading to hyperlipidemia [28]. Over the past few years, PCSK9 inhibitors have become a new hot spot for drug development in hyperlipidemia and atherosclerotic heart disease. To elucidate whether PCSK9 was implicated in mTOR inhibitors-induced hyperlipidemia, situ breast cancer models in BALB/c mice were established. It was found that PRS could overcome the effects of LDLR down-regulation during tumor therapy. As shown in Figure 5A,B, the results were consistent with the findings of AI D. et al. [28]. In addition, when compared with the Blank group, LDLR levels were significantly lower in the R + S group. This might be due to the non-specific distribution of free drugs. Interestingly, when compared with the other groups, the LDLR levels of the PRS group were significantly increased. This indicated that targeted co-delivery of PCSK9 inhibitors might, on the one hand, increase the targeting of Rapamycin to tumors by the carrier, weakening its effect on liver LDLR. On the other hand, the expression of LDLR was increased by the pharmacological effects of the PCSK9 inhibitor, which avoided abnormalities in lipid metabolism caused by non-specific targeting and distribution of Rapamycin. Nevertheless, the blood lipids of each group of mice were not detected in our study. In conclusion, the mechanism of PCSK9 inhibitors on Rapamycin-induced hyperlipidemia needs to be further studied.

## 5. Conclusions

In summary, a chimeric peptide engineered nanomedicine (called PRS) was constructed via the self-assembly of mTOR inhibitor, PCSK9 inhibitor, and palmitic-K(palmitic)CREKA. It demonstrated that palmitic-K(palmitic)CREKA enhanced the stability and tumor targeting ability of Rapamycin and SBC-115076 for improved drug delivery efficiency. PRS also acquired all of the bioactivity of the assembled drugs, including good pro-apoptosis and LDLR regulation. More importantly, because PRS had better cell uptake and tumor retention properties, it showed a higher anti-tumor effect. In vivo and in vitro experiments showed that PRS could significantly improve the effectiveness of tumor treatment and had good biosafety. In addition, PRS could regulate the expression of LDLR protein levels in the liver of 4T1 tumor-bearing mice. This study provided a new idea for co-delivery drugs in conjunction with anti-tumor therapy and therapy-induced hyperlipidemia.

## Data Availability

Data are contained within the article or Appendix A.

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
