# Peer review of "Chimeric Peptide Engineered Nanomedicine for Synergistic Suppression of Tumor Growth and Therapy-Induced Hyperlipidemia by mTOR and PCSK9 Inhibition"

_pharmaceutics, 2023, doi:10.3390/pharmaceutics15102377_

Round 1
Reviewer 1 Report
Reducing the side effects induced by chemotherapy is critical to the health and welfare of the patients. To do this, the authors designed a novel experiment with chimeric peptide-engineered nanomedicine showing synergistic suppression of tumor growth and therapy-induced hyperlipidemia by mTOR and PCSK9 inhibition. The experiments were well-designed. The presented results supported their hypothesis that co-delivery of PCSK9 and mTOR inhibitors may provide a promising strategy to avoid tumor chemotherapy-induced hyperlipidemia. Although this work is in the initial stage, it will be an interesting strategy, potentially beneficial to the patient. As a reviewer and reader, I recommend this work be accepted for publication in Pharmaceutics.
Author Response
Dear Professor,
Thanks very much for taking your time to review this manuscript. Thank you very much for your affirmation and approval to our article. Thank you again for your positive comments and valuable suggestions to improve the quality of our manuscript.
Best regards!

Reviewer 2 Report
In this manuscript, the authors report chimeric peptide engineered nanomedicine (PRS) for synergistic supression of tumor-growth and undesired hyperlipidemia. The authors fabricated PRS by assembling lipid-grafted chimeric peptide with mTOR inhibitor (rapamycin) and Pcsk9 inhibitor (SBC-115076). Overall, the manuscript elaborately demonstrated the antitumor effects of PRS. However, there are few minor concerns in experimental results. Therefore, I would like to recommend to accepting this manuscript to Pharmaceutics after major revision.
1) In Figure 4F, histological images of the lungs appear to show significant changes in the peptide-administered group, but the authors claim that there are no significant changes. Likewise, serum ALT level (in Figure 4A) seems to be elevated significantly.
2) In Figures 3C, D, the tumor volume of P, PR, PS, R+S groups and PRS group seems to be not significantly different.
3) In Figure 2, only R and only S-treated group should be considered as important control groups. However, these are totally missing.
Author Response
|
Dear Professor, We sincerely appreciate the time and effort that you dedicated to providing the insightful comments on and valuable improvements to our paper. According to the comments, we have made extensive modifications to our manuscript and supplemented extra reference to make our results convincing. In this revised version, changes to our manuscript were all highlighted within the document by using red-colored text. Point-by-point responses to the nice reviewer is listed below this letter. Best regards! |
|
Comments 1: 1) In Figure 4F, histological images of the lungs appear to show significant changes in the peptide-administered group, but the authors claim that there are no significant changes. Likewise, serum ALT level (in Figure 4A) seems to be elevated significantly. |
|
Response 1: We agree that there is an excellent consideration. Firstly, we have added two references to support this idea as follows. 32. Chu, R.; Wang, N.; Bi, Y.; Nan, G. Rapamycin prevents lung injury related to acute spinal cord injury in rats. Sci Rep. 2023, 13, 10674. [CrossRef] [PubMed] 33. Ye, P.; Jiang, X. M.; Qian, W. C.; Zhang, J. Inhibition of PCSK9 improves the development of pulmonary arterial hypertension via down-regulating notch 3 expression. Cardiovasc Drugs Ther. 2023. [CrossRef] [PubMed] According to Reference 32, as shown in Figure 2, after the treatment with Rapamycin, infiltration of inflammatory cells in lungs could be reduced. According to Reference 33, PCSK9 inhibitor could reduce the pulmonary vascular remodeling in Figure 3. Therefore, we have supplemented the “Meanwhile, as shown in Figure 4F, H&E staining of major organ tissues such as heart, liver, spleen and kidneys did not show significantly histological abnormalities. However, compared with R + S and PRS groups, Blank, P, PR, PS groups showed infiltration of inflammatory cells, resulting in lungs structural damage. The reason might be that the combination Rapamycin and PCSK9 inhibitor could be protective against the effects of pulmonary injury. Ruiliang Chu and his colleagues had found that infiltration of inflammatory cells and hemorrhage in the lung tissue was alleviated after the treatment of Rapamycin [32]. It was observed that PCSK9 monoclonal antibody could reduce hypoxia-induced pulmonary vascular remodeling, which was associated with the inhibition of inflammation [33]”. Mention exactly where in the revised manuscript this change can be found – page 12-13, and line 333-342. Meanwhile, thank you for pointing out the problem of the serum ALT level. Although serum ALT level (in Figure 4A) seems to be elevated significantly, the arithmetic mean of serum ALT level was in the normal range. |
|
Comments 2: In Figures 3C, D, the tumor volume of P, PR, PS, R+S groups and PRS group seems to be not significantly different. |
|
Response 2: We agree that this is an important consideration, it is actually that the tumor volume of P, PR, PS, R + S groups and PRS group seems to be not significantly different in Figure 3C, D, but the tumor weight of P, PR, PS, R + S groups and PRS group had shown significantly different in Figure 3E. Therefore, we considered that compared with other control groups, PRS group significantly showed inhibition of tumor growth.
Comments 3: In Figure 2, only R and only S-treated group should be considered as important control groups. However, these are totally missing. Response 3: Thank you for your reminder. We agree with this comment. Therefore, we have supplemented the “It was displayed that PRS group had significantly high cytotoxicity, but only R and only S-treated groups (like PR, PS, R + S groups) were shown low cytotoxicity. Of which, it could be link to Rapamycin and SBC-115076 two drugs for synergy”. Mention exactly where in the revised manuscript this change can be found – page seven, paragraph four, and line 262-265. |

Reviewer 3 Report
The property of rapamycin to inhibit cancer was discovered relatively recently, classically it is used as an immunosuppressant in transplants. In this regard, research on the effectiveness of its use for the treatment of various types of cancer is extremely topical and important. The authors developed a molecular engineering structure containing active components mTOR inhibitor and PCSK9 inhibitor and tested it on cellular and animal models.
To assess the biosafety of the developed system, authors also demonstrated that the main blood parameters do not change with the injection of nanoparticles. Unfortunately, I didn’t find information on the biochemical analyzes performed (AST, ALT, Urea, UA) in the materials and methods section. This information should be added.
Besides, there is no information on the procedure of Cy5.5 labeling, it should be added.
Author Response
|
Dear Professor, We feel great thanks for your professional review work on our article. As you are concerned, there are several problems that need to be reviewed. According to your nice suggestions, we have made some corrections to our previous draft, the detailed corrections are listed below. Point-by-point responses to the nice reviewer is listed below this letter. Best regards! |
|
Comments 1: To assess the biosafety of the developed system, authors also demonstrated that the main blood parameters do not change with the injection of nanoparticles. Unfortunately, I didn’t find information on the biochemical analyzes performed (AST, ALT, Urea, UA) in the materials and methods section. This information should be added. |
|
Response 1: We agree that there are wonderful considerations. We feel sorry for our carelessness. Therefore, we have supplemented the” Blood biochemical tests was conducted by automated chemistry analyzer (Chemray-240)”and “The venous blood sample was collected from the external jugular vein. Biochemical analysis included aspartate aminotransferase (AST), alanine transaminase (ALT), UREA and uric acid (UA), which were determined using automated chemistry analyzer. Regular quality control procedures were followed and the analyzers were managed in accordance with the manufacturers' instructions”. Mention exactly where in the revised manuscript this change can be found – page 4 and line 118-119 and page 6 and line 209-213. |
|
Comments 2: Besides, there is no information on the procedure of Cy5.5 labeling, it should be added. Response 2: Thank you for your kindness reminder. We agree with this comment. Therefore, As suggested by the reviewer, we have supplemented the “Cy5.5-PRS was prepared as follow. Rapamycin (10 mg/mL), SBC-115076 (10 mg/mL), and Cy5.5 (5 mg/mL) in DMSO were dispersed into ultra-pure water and disposed with ultrasonication for 50 seconds. Afterwards, 25 mg/mL of palmitic-K(palmitic)CREKA was added to the above solution and the mixture was stirred with ultrasonication for 5 minutes. Then, resultant mixtures were on dialysis (MWCO 1000 Da) to eliminate the rest of free drugs”. Mention exactly where in the revised manuscript this change can be found – page 4 and line 144-149. |

Round 2
Reviewer 2 Report
The authors revised the comments point by point.